# Severe infections in peritoneal dialysis and home hemodialysis patients: An inception cohort study

**Wisam Bitar**[1]*, **Jaakko Helve**[1,2], **Mari Kanerva**[3], **Eero Honkanen**[1], **Virpi Rauta**[1,4], **Mikko Haapio**[1], **Patrik Finne**[1,2]

**1** Department of Nephrology, University of Helsinki and Helsinki University Central Hospital, Helsinki, Finland, **2** Finnish Registry for Kidney Diseases, Finnish Kidney and Liver Association, Helsinki, Finland, **3** Department of Infectious Diseases, University of Helsinki and Helsinki University Central Hospital, Helsinki, Finland, **4** IT Management, Helsinki and Uusimaa Hospital District, Helsinki, Finland

\* wisam.bitar@hus.fi, wisambr@hotmail.com

## Abstract

### Objectives

Infections are the most common non-cardiovascular cause of death among dialysis patients. Earlier studies have shown similar or higher risk of infectious complications in peritoneal dialysis (PD) compared to hemodialysis (HD) patients, but comparisons to home HD patients have been rare. We investigated the risk of severe infections after start of continuous ambulatory PD (CAPD) and automated PD (APD) as compared to home HD.

### Methods

All adult patients (n = 536), who were on home dialysis at day 90 from starting kidney replacement therapy (KRT) between 2004 and 2017 in Helsinki healthcare district, were included. We defined severe infection as an infection with C-reactive protein of 100 mg/l or higher. Cumulative incidence of first severe infection was assessed considering death as a competing risk. Hazard ratios were estimated using Cox regression with propensity score adjustment.

### Results

The risk of getting a severe infection during the first year of dialysis was 35% for CAPD, 25% for APD and 11% for home HD patients. During five years of follow-up, the hazard ratio of severe infection was 2.8 [95% CI 1.6–4.8] for CAPD and 2.2 [95% CI 1.4–3.5] for APD in comparison to home HD. Incidence rate of severe infections per 1000 patient-years was 537 for CAPD, 371 for APD, and 197 for home HD patients. When excluding peritonitis, the incidence rate was not higher among PD than home HD patients.

**Data Availability Statement:** The data protection law of Finland (5.12.2018/1050), which is based on the General Data Protection Regulation (GDPR) of EU, does not allow public (or any) sharing of

individual patients' data. Our research permission granted by Helsinki University Hospital does not either allow sharing of individual patients' data. Researchers can submit a research application to the Helsinki University Hospital in order to get access to the data. The contact person: Professor, chief physician Pauli Puolakkainen, Helsinki University Hospital, email: Pauli. puolakkainen@hus.fi.

**Funding:** This study was supported by Finska läkaresällskapet, and Liv och Hälsa. The funders had no role in study design, data collection and analysis, decision to publish, or preparation of the manuscript.

**Competing interests:** I have read the journal's policy and the authors of this manuscript have the following competing interests: Dr. Finne has received honoraria for lectures from Baxter, AstraZeneca and Boehringer-Ingelheim, and he is a member in an advisory board of Baxter. Dr. Rauta has received honoraria for lectures from Baxter, Fresenius and AstraZeneca, and research funding (not for this project) from Business Finland; and she is a board member in Finnish Society of Nephrology. Dr. Honkanen has received honoraria for lectures from AstraZeneca and Fresenius Medical Care. The findings, results, and conclusions in this report are those of the authors and are independent from the funding sources.

## Conclusions

CAPD and APD patients had higher risk of severe infections than home HD patients. This was explained by PD-associated peritonitis.

## Introduction

Infections are the most common cause of death among dialysis patients next to cardiovascular disease [1, 2]. Dialysis patients have been reported to carry an 82-fold risk of dying of infections compared to the general population [3]. Earlier studies have shown that patients on peritoneal dialysis (PD) and hemodialysis (HD) have similar risks of infection [4–6], whereas one study showed increased risk of infection-related hospitalization among PD compared to HD patients [7]. When comparing patients on continuous ambulatory PD (CAPD) and automated PD (APD) some studies have not found a difference in risk of infections [8–11], while others have indicated higher risk in CAPD than APD patients [12–14]. Infections have been reported to be the most common cause of hospital admissions among home HD patients [15]. Two studies have compared infection-related hospitalization between PD and home HD, and they have given contradicting results. The first one demonstrated no difference between the treatment modalities in an intention-to-treat analysis [16], whereas the other one showed that patients who entered PD had higher risk of infection-related hospitalization than those who entered home HD [17]. Neither study considered CAPD and APD separately.

For patients who are eligible for home dialysis, there are three options: CAPD, APD and home HD. We have recently reported similar survival of patients on APD and home HD [18], whereas earlier studies have reported somewhat less favorable survival in PD compared to home HD [16, 19–24]. However, when considering which type of home dialysis to choose, other issues than survival are important as well. While peritonitis is a well-known problem in PD, there is scarce and contradicting information about the difference in infection-burden between PD and home HD. We investigated the risk of severe infections after start of CAPD or APD as compared to home HD.

## Materials and methods

### Study design and population

The Helsinki-Uusimaa healthcare district covers 1.7 million inhabitants, representing about 30% of Finland's population. In this healthcare district, 1747 adult patients started dialysis treatment between 2004 and 2017. Of these, 536 patients were on home dialysis at day 90 from initiating kidney replacement therapy (KRT) and they were included in this study. Patients' follow-up data were available until December 31, 2019.

When the dialysis modality is selected, our healthcare district applies a "home first"-policy, with equal priority given to PD and home HD. The patients are informed using on-spot and video information, and they are evaluated by a multidisciplinary team for home dialysis suitability. According to surveillance data from our healthcare district, the frequency of PD peritonitis was 530/1000 patient-years in 2011–2020 which is slightly above the target of less than 500 PD peritonitis episodes/1000 patient-years according to the ISPD guideline 2016 [25] and clearly above the newest target of 400/1000 patient-years in the ISPD guideline 2022 [26]. Our unit has a protocol for peritonitis prevention that includes education of patients by specialized PD nurses, exit-site care, surveillance of PD peritonitis frequency and regular follow-up meetings with infection diseases specialists. All patients who start PD are trained by a PD nurse for

at least one week, and retraining is provided if needed. Prophylactic antibiotics are used at PD catheter insertion and prior to dental and endoscopic procedures. Decolonization is performed after Staphylococcus aureus infection, and topical prophylactic antibiotic is used on exit-site based on clinical assessment. All home HD patients were trained during an average of six weeks to perform hemodialysis at home and all patients used the button-hole technique for cannulation. On average, at the end of the years 2004 to 2019, 91% of the home HD patients had a fistula, 4% had a graft and 5% had a tunneled hemodialysis catheter.

As the study subjects were not contacted and no intervention was performed, no consent from the patients nor permission from an Ethics committee were needed. The study was approved by the Helsinki University Hospital, permission number HUS/459/2018.

### Data collection

A structured research database was established, and extensive data were extracted from the patient data system. The retrospectively collected data comprised 34 various comorbidities, primary renal disease (PRD), laboratory results, blood pressure, ECG parameters, findings on heart ultrasound, and social variables (S1 Table). Additionally, data on all values of plasma C-reactive protein (CRP) of the study patients were retrieved from the data pool of the Helsinki-Uusimaa healthcare district. In Finnish medical practice, CRP has for decades been used as a key infection marker for the assessment of infection severity and bacterial origin, and its predictability is often appreciated over the blood leucocyte count. CRP is also used to guide antibiotic therapy. Therefore, several measurements per patient are usually available during an infection episode.

### Identification of severe infections

We defined severe infection as an infection accompanied with a plasma CRP level of 100 mg/l or higher. Although several studies have shown that CRP correlates with severity of infections [27, 28], there is no general recommendation in the literature for the cutoff of CRP to determine the severity. We chose the cutoff of 100 mg/l because a CRP value under this level has been strongly associated with low infection mortality [29] and among HD patients a CRP value higher than 100 mg/l has been correlated with risk of infection mortality [30]. In our study, a severe infection episode was recorded when CRP was 100 mg/l or higher, and the cause was an infection. Two different episodes were recorded if the time interval between elevated CRP values was at least 21 days without CRP measurements, or 30 days with at least one CRP value less than 50 mg/l in between. The cause of elevated CRP was checked from the patient's medical records and defined as infectious or non-infectious. The cause of infection, including types of infections and infectious agents, was collected. For PD patients, peritonitis and PD-catheter related infections were defined as dialysis-related. Bloodstream infection was considered dialysis-related only if accompanied by peritonitis or PD access site infection. For home HD patients, bloodstream infections were considered dialysis-related if the source was clearly related to the dialysis access site infection or if it was unknown (i.e., not related to other infection foci such as urinary tract infections). We also collected data on hospitalization, length of hospitalization and need for intensive care in connection to the infection episode. Infections were defined as fatal if the patient died within 30 days from the infection regardless of the cause of death.

### Statistical analyses

The probability of the first infection episode was estimated considering death as a competing risk event according to type of home dialysis treatment (CAPD, APD or home HD) at day 90

from the beginning of KRT. Patients were followed up for five years from the beginning of KRT until kidney transplantation (n = 285), transfer to in-center HD (n = 122, of whom 58 were on CAPD, 49 on APD and 15 on home HD, and 35% of the PD patients had transferred because of peritonitis), change from CAPD or APD to home HD (n = 4), change from home HD to CAPD or APD (n = 1), transfer to conservative treatment (n = 2) loss of follow up (n = 1), death (n = 71) or end of follow-up at 5 years or 31 December 2019 (n = 50). Hazard ratios of first infection episode associated with each home dialysis modality were assessed using Cox regression. The event was the first infection episode and patients were censored at end of follow-up. As the research question was aetiological (to study association between dialysis modality and infection risk) we selected Cox regression without taking competing risk into account [31]. We further did sensitivity analyses using Fine-Gray regression in which 1) death, 2) both death and kidney transplantation or 3) death, kidney transplantation and transfer to in-center HD were defined as competing risk events. In all analyses we also adjusted for propensity scores [32, 33] to account for a possible confounding effect of age, comorbidities, primary renal disease, and other factors (S1 Table). The proportional hazards assumption was assessed graphically by inspecting the cumulative hazards of first infections episodes among CAPD, APD and home HD patients. In addition, the proportional hazards assumption was tested statistically (using the estat phtest) in the Stata software Version 17.

We imputed missing values using predictors as indicated in S1 Table. In addition, waitlisting for kidney transplantation, kidney transplantation, and death were included as predictors in the imputation. Each comorbidity was analyzed as either found or not found in the patient files before the beginning of KRT. Consequently, there were no missing data for comorbidities.

We used binary logistic regression to develop propensity scores for CAPD vs. home HD, APD vs. home HD and PD (CAPD and APD as one group) vs. home HD pairwise. S1 Table displays all the variables from which the explanatory variables for the propensity scores were selected by a stepwise forward procedure. The *P* value for inclusion was less than 0.1. As a sensitivity analysis we adjusted, on top of the propensity score adjustment, for prognostic variables, which predicted infections but were not included in the propensity scores. These variables were selected from the variables in S1 Table using stepwise forward Cox regression. Four statistically significant prognostic variables were identified: plasma phosphate, history of coronary intervention, stroke, and vasculitis.

Incidence rate (IR) and IR ratios (IRR) of infections were assessed for CAPD, APD and home HD patients. Confidence intervals (95% CI) of IRR were calculated [34]. The cumulative incidence was estimated using the R Software (version Ri386 3.6.0, the cmprsk package, cuminc function), while all other statistical analyses were performed using SPSS (version 25).

## Results

### Population characteristics

Of the patients who were on home dialysis modality at day 90 from starting KRT, 162 (30%) were on CAPD, 229 (43%) on APD and 145 (27%) on home HD (Table 1). Compared to home HD patients, CAPD patients were older and had a larger number of comorbidities, whereas the characteristics of APD patients were similar to those of home HD patients. CAPD patients more often needed dialysis assistance and they were less often waitlisted for kidney transplantation. At the time of start of dialysis, the median plasma CRP was 4 mg/l, with an interquartile range of 3–7 mg/l and a 90-percentile of 15 mg/l, which means that 90% of the patients had a baseline CRP value lower than 15 mg/l.

**Table 1. Patient characteristics.**

| Home dialysis modality | CAPD | APD | Home HD | P-value | |
|---|---|---|---|---|---|
| | | | | CAPD vs Home HD | APD vs Home HD |
| Number of patients | 162 | 229 | 145 | | |
| Male Gender (%) | 70 | 66 | 68 | 0.691 | 0.703 |
| Median FU-time (years) | 2.0 (0.97–3.4) | 1.6 (0.86–2.5) | 1.4 (0.73–2.7) | 0.013 | 0.499 |
| Number of Patient-years | 365 | 426 | 264 | | |
| Number of Infection episodes | 197 | 158 | 52 | | |
| Hospitalization needed (n) | 178 | 142 | 38 | | |
| Infection episode IR | 540 | 371 | 197 | | |
| At least one episode (%) | 62 | 35 | 21 | | |
| Number of deaths in 5 years* | 66 | 26 | 12 | <0.001 | 0.330 |
| Number of infection deaths* | 18 | 12 | 5 | 0.011 | 0.418 |
| Median age (years) | 65 (52–74) | 50 (40–61) | 50 (42–60) | <0.001 | 0.832 |
| Primary renal disease (%) | | | | <0.001 | 0.017 |
| Glomerulonephritis | 17 | 22 | 24 | | |
| Cystic kidney disease | 6 | 15 | 31 | | |
| Type 1 diabetes | 19 | 21 | 17 | | |
| Type 2 diabetes | 20 | 9 | 5 | | |
| Interstitial nephritis | 4 | 4 | 1 | | |
| Hypertension | 5 | 4 | 2 | | |
| Unknown | 20 | 13 | 10 | | |
| Others | 11 | 12 | 9 | | |
| Number of comorbidities (%) | | | | 0.001 | 0.305 |
| 0 | 1.2 | 2.2 | 0.0 | | |
| 1–2 | 32 | 44 | 48 | | |
| 3–4 | 40 | 42 | 41 | | |
| > = 5 | 27 | 12 | 11 | | |
| Dialysis assistance (%) | | | | <0.001 | 0.061 |
| by professional | 4.9 | 1.7 | 0.7 | | |
| by family member | 18.5 | 4.4 | 4.1 | | |
| KTx-listed (%) * | 39 | 75 | 74 | <0.001 | 0.776 |
| KTx (%) * | 25 | 58 | 66 | <0.001 | 0.116 |

Abbreviations: HD, hemodialysis; PD, peritoneal dialysis; APD, automated PD; CAPD, continuous ambulatory PD; (%), (percentage); FU, follow-up; IQR, interquartile range; IR, incidence rate per 1000 patient-years; KTx, kidney transplantation;

*During the follow-up period.

## Cumulative incidence and hazard ratio of the first severe infection episode

Of the 536 patients, 213 (40%) experienced at least one severe infection episode during the follow-up. Cumulative incidence of a first infection episode was 35% for CAPD, 25% for APD and 11% for home HD patients at one year from start of dialysis (Fig 1) (P < 0.001). Both in unadjusted and in age- and sex-adjusted analyses, CAPD and APD patients had a 2–3-fold risk of a first infection episode compared to home HD patients (Table 2). Similarly, after adjustment for propensity scores, the hazard ratio of a first infection episode was 2.8 (95% CI 1.6–4.8) for CAPD patients and 2.2 (95% CI 1.4–3.5) for APD patients compared to home HD patients. We tested statistically that the proportional hazards' assumption was not violated (P = 0.469). We further performed four sensitivity analyses in which, instead of censoring, we

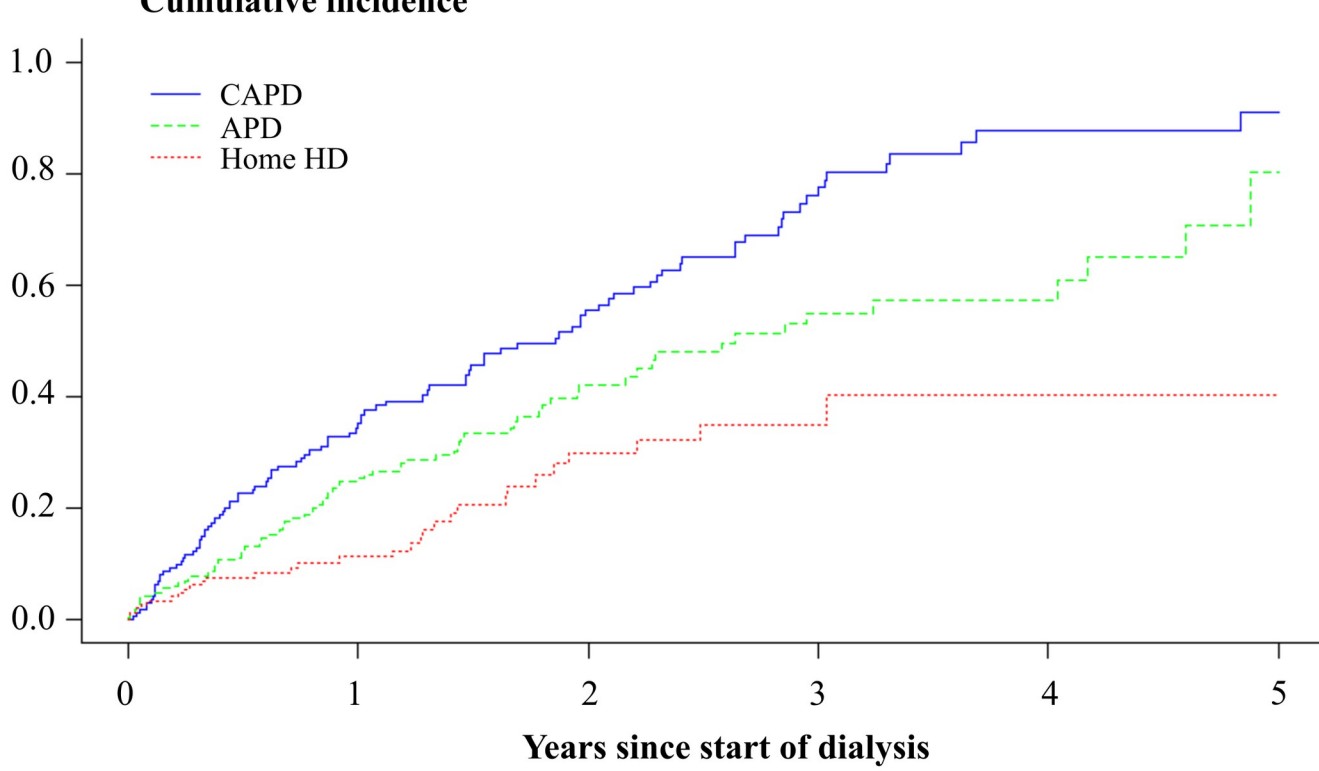

**Fig 1. Cumulative incidence of the first severe infection episodes.**

defined 1) death, 2) both death and kidney transplantation or 3) death, kidney transplantation and transfer to in-center HD as competing risk events in Fine-Gray regression, but the hazard ratios remained virtually unchanged. In the fourth sensitivity analysis, we adjusted for prognostic variables in addition to the propensity scores, but the results remained virtually

**Table 2. Hazard ratios of first infection according to home dialysis modality during 5 years from the start of KRT.**

| | HR of first infection episode | 95% CI of HR | |
|---|---|---|---|
| | | Lower | Upper |
| HR of first infection episode incidence, unadjusted | | | |
| Home HD (reference) | | | |
| CAPD | 3.2 | 2.1 | 4.8 |
| APD | 1.9 | 1.2 | 2.8 |
| HR of first infection episode incidence, adjusted for age and sex | | | |
| Home HD (reference) | | | |
| CAPD | 2.7 | 1.7 | 4.1 |
| APD | 1.9 | 1.2 | 2.8 |
| HR of first infection episode incidence, adjusted for propensity score | | | |
| CAPD (compared to Home HD) | 2.8 | 1.6 | 4.8 |
| APD (compared to Home HD) | 2.2 | 1.4 | 3.5 |

Abbreviations: KRT, kidney replacement therapy; HR, hazard ratio; CI, confidence interval; HD, hemodialysis; PD, peritoneal dialysis; APD, automated PD; CAPD, continuous ambulatory PD.

unchanged: hazard ratio 2.6 (95% CI 1.5–4.5) for CAPD patients and 2.0 (95% CI 1.3–3.3) for APD patients compared to home HD patients. We also did an additional analysis comparing the risk of a first infection episode between APD and CAPD and found that the difference was not significant after adjusting for propensity scores, HR 1.3 (95% CI 0.94–1.9).

## Incidence rate of severe infection episodes

Calculation of the incidence rate allows considering all severe infection episodes, not only the first infection of each patient. The incidence rate was 537 infection episodes per 1000 patient-years for CAPD patients while it was 371 for APD and 197 for home HD patients (Table 3). The incidence rate ratio (IRR) was 2.7 (95% CI 2.0–3.7) for CAPD and 1.9 (95% CI 1.4–2.6) for APD compared to home HD patients. If excluding dialysis-related infections, the corresponding IRR was 0.90 (95% CI 0.65–1.2) for CAPD and 0.65 (95% CI 0.47–0.89) for APD. On the other hand, if analyzing only the dialysis-related infections, CAPD patients had an IRR of 14 (95% CI 6.7–31) and APD patients an IRR of 9.7 (95% CI 4.5–21) compared to home HD patients. Of all the 407 severe infection episodes, 5% were fatal, i.e., the patient died within 30 days.

## Causes of severe infection episodes

Peritonitis constituted 68% of the infection episodes among CAPD patients (incidence rate 367 per 1000 patient-years) and 67% of the episodes among APD patients (incidence rate 249 per 1000 patient-years) (Table 3). Bloodstream infections accounted for 21% of the infection episodes among home HD patients, but only 3% of the CAPD and APD patients' episodes. The IRR of all bloodstream infections was 0.39 (95% CI 0.21–0.73) for CAPD and 0.28 (95% CI 0.15–0.52) for APD patients compared to home HD patients.

**Table 3. Type and incidence rate of infections according to type of home dialysis modality.**

| Type and incidence rate of infections | | | | |
|---|---|---|---|---|
| IR per 1000 patient-years (number of infections) | | | | |
| Home dialysis modality | | CAPD | APD | Home HD |
| Number of patient-years | | 365 | 426 | 264 |
| IR of all infections (n) | | 540 (197) | 371 (158) | 197 (52) |
| IRR of all infections (95% CI) | | 2.7 (2.0–3.7) | 1.9 (1.4–2.6) | 1 (reference) |
| IR of dialysis- related infections (n) | Peritonitis | 367 (134) | 249 (106) | 0 (0) |
| | Access related | 14 (5) | 9.4 (4) | 8 (2) |
| | Bloodstream infection | 2.7 (1) | 0 (0) | 19 (5) |
| IR of dialysis-related infections (n) | | 383 (140) | 258 (110) | 27 (7) |
| IRR of dialysis-related infections (95%CI) | | 14 (6.7–31) | 9.7 (4.5–21) | 1 (reference) |
| IR of all bloodstream infections (n) | | 16 (7) | 12 (5) | 42 (11) |
| IRR of all bloodstream infections (95% CI) | | 0.46 (0.25–0.85) | 0.28 (0.15–0.52) | 1 (reference) |
| IR of non-dialysis-related infections (n) | | 156 (57) | 113 (48) | 170 (45) |
| IRR of non-dialysis-related infections (95% CI) | | 0.90 (0.65–1.2) | 0.65 (0.47–0.89) | 1 (reference) |
| IR of infections needed hospitalization (n) | | 488 (178) | 333 (142) | 144 (38) |
| IRR of infections needed hospitalization (95% CI) | | 2.5 (1.8–3.4) | 1.7 (1.2–2.3) | 1 (reference) |
| IR of fatal infections (n) | | 41 (15) | 9 (4) | 4 (1) |
| IRR of fatal infections (95% CI) | | 11 (1.5–80) | 2.5 (0.33–18) | 1 (reference) |

Abbreviations: IR, incidence rate; n, number of infections; PD, peritoneal dialysis; CAPD, continuous ambulatory PD; APD, automated PD; HD, hemodialysis; IRR, incidence rate ratio; CI, confidence interval.

**Table 4. Types of non-dialysis-related infections.**

| Infection Type | CAPD | APD | Home HD | Total |
|---|---|---|---|---|
| Upper or lower respiratory airways | 15 | 13 | 5 | 33 |
| Skin or soft tissue | 12 | 11 | 11 | 34 |
| Gastrointestinal or hepatobiliary | 3 | 1 | 10 | 14 |
| Bone or Joint | 8 | 3 | 1 | 12 |
| Genitourinary | 5 | 9 | 4 | 18 |
| Bloodstream infection | 6 | 5 | 6 | 17 |
| Unknown origin | 8 | 6 | 8 | 22 |
| Total | 57 | 48 | 45 | 150 |

Abbreviations: PD, peritoneal dialysis; CAPD, continuous ambulatory PD; APD, automated PD; HD, hemodialysis.

Of the 150 non-dialysis-related infection episodes 34 (23%) were caused by skin or soft tissue infections and 33 (22%) by respiratory infections (Table 4).

The most common causative organism of peritonitis was Staphylococcus Aureus (24%), and the second most common cause Coagulase-negative Staphylococcus (17%) (Table 5).

## Hospitalization in connection to severe infection episodes

Of all the 407 severe infection episodes, 358 (88%) led to hospitalization. Among CAPD and APD patients 90% (320 of 355) and among home HD patients 73% (38 of 52) of the episodes required hospitalization (P < 0.001, Table 1). Overall, the median duration of hospital stay was 6 days for CAPD and APD patients and 8 days for home HD patients (P = 0.25). Regardless of CRP level, hospitalization was needed in 86–90% of the infection episodes. However, CRP level associated with the proportion of infection episodes requiring more than 7 days of hospital care: 32% if CRP level was 100–149 mg/L, 43% if 150–199 mg/L and 60% if 200 mg/L or higher (P < 0.001). Intensive care was needed in 2.2% of all the 407 infection episodes, in 4.9% of the 103 episodes with a CRP of 200 mg/l or higher and in 1.3% of the 304 episodes with a lower CRP level (P = 0.049). Among PD patients, 98% of the infection episodes caused by peritonitis and 75% of the non-peritonitis infection episodes required hospitalization. Of the 240 infection episodes caused by peritonitis, PD-catheter was removed in 12%, and this proportion increased with increasing CRP level (P = 0.040).

**Table 5. Causative organisms of the peritonitis episodes.**

| Causative organism | CAPD | APD | Total (%) |
|---|---|---|---|
| Staphylococcus Aureus | 33 | 25 | 58 (24) |
| Coagulase-negative Staphylococcus | 27 | 14 | 41 (17) |
| Escherichia coli | 8 | 2 | 10 (4) |
| Pseudomonas | 11 | 6 | 17 (7) |
| Fungi | 1 | 5 | 6 (3) |
| Culture negative | 13 | 18 | 31 (13) |
| Other agents or multiple microbial agents | 41 | 36 | 77 (32) |
| Total | 134 | 106 | 240 (100) |

Abbreviations: PD, peritoneal dialysis; CAPD, continuous ambulatory PD; APD, automated PD.

### Severe infection episodes before start of dialysis

Both in PD patients and home HD patients the incidence rate of infections was clearly higher after the start of KRT than before (Fig 2). Among PD patients a slight increase in incidence rate could be noticed already half a year prior to start of dialysis. However, if analyzing the whole two-year period before start of KRT, infection rate did not differ significantly between groups with an IRR of 1.5 (95% CI 0.89–2.7) for CAPD and 1.2 (95% CI 0.68–2.0) for APD patients in comparison to home HD patients.

## Discussion

In this study of patients who started home dialysis, we found that CAPD and APD patients have a 2–3-fold risk of getting severe infections compared to home HD patients. This was mainly explained by the numerous peritonitis episodes occurring among PD patients. Infections that were unrelated to dialysis were not more frequent among PD than home HD patients, and PD patients had lower risk of bloodstream infection. Although infections that raised plasma CRP to a level of 100 mg/L or higher were frequent and occurred among 40% of the patients, only 5% died within 30 days from an infection episode.

There is limited information about the infection burden among home HD patients compared to PD patients, and no study has compared home HD to CAPD and APD separately. However, comparisons have been made between CAPD and APD, showing contradicting results. Some studies showed a higher incidence of peritonitis among CAPD than APD patients [12–14], whereas others did not [8–11]. Although this was not the primary comparison in our study, we observed a tendency towards higher risk of peritonitis in CAPD than APD patients. A few studies have shown similar or increased infection burden among PD compared to in-center HD patients [4–7].

**Incidence of infection episodes per 1000 patient-years**

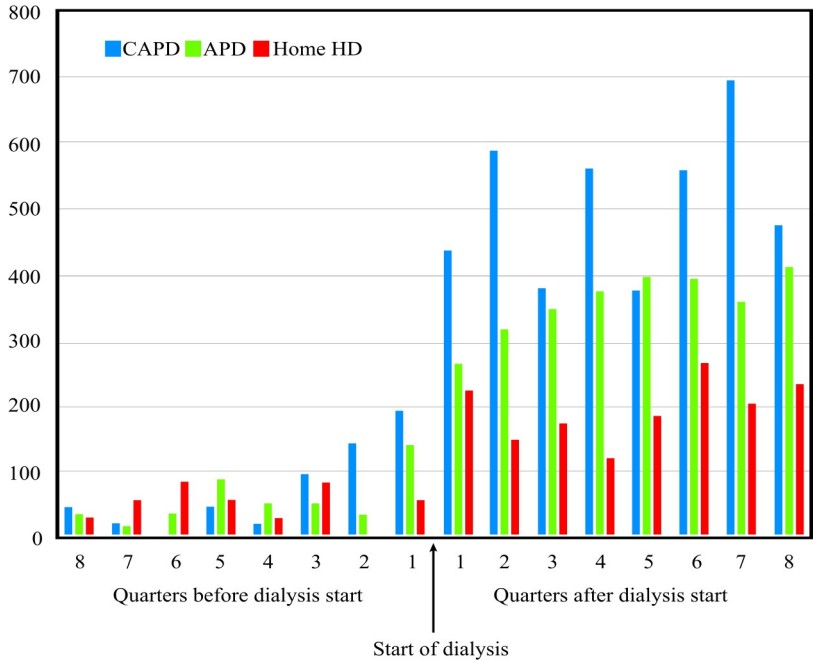

**Fig 2. Incidence of severe infection episodes per 1000 patient-years quarter-yearly before and after the start of dialysis.**

Rydell et al. [17] studied hospitalizations among 152 home HD patients compared to 456 matched PD patients. Home HD patients had a lower infection-related hospitalization rate, which is in line with our results on severe infections. However, Rydell et al. did not present type of infections or to what extent this difference was due to peritonitis. Weinhandl et al. [16] compared US home HD patients with matched PD patients, and in intention-to-treat analysis they did not find differences in overall risk of infection-related hospitalizations between the groups. Home HD patients had higher risk of bloodstream infection and lower risk of dialysis-access site infection including peritonitis, which is comparable to our results.

The strength of this population-based study was that all patients in the healthcare district who were on home dialysis at day 90 from the start of KRT were included, thus minimizing selection bias. As we had data on all patients' CRP values, the data on infection episodes with a CRP of 100 mg/l or higher were complete and assessed in the same way regardless of dialysis modality. A CRP value of 100 mg/l or higher has not been validated as a marker of severe infections. Notably, our results indicate that CRP level associates with infection severity with regard to longer hospitalization and need for intensive care, and in case of peritonitis, with removal of PD catheter. Thus, identifying severe infections based on CRP value appears sensible. Another strength is that we had extensive data on putative confounding factors, which were used for propensity score adjustment when comparing the dialysis modalities. Furthermore, the main characteristics of patients on APD and home HD were similar, which should reduce bias in this comparison. Nevertheless, in observational studies like this, there is still a possibility of residual confounding. Furthermore, the distribution of patients in various dialysis modalities differs between countries and centers, and the proportion of home HD patients is often lower than in this study, which may limit the generalizability of our results. On the other hand, the large proportion of home HD patients in our study implies a less strict selection of patient to this modality, and may improve comparability with PD. We reported a high frequency of hospitalization for peritonitis due to the policy of our unit to start most peritonitis treatments in the hospital. Thus, hospitalization rates for peritonitis cannot be generalized to units with other policies. Notably, our analysis included only peritonitis episodes accompanied with a plasma CRP of 100 mg/l or higher. If not restricting to elevated CRP, the peritonitis rate is higher, i.e., it was 530 episodes per 1000 patient-years in 2011–2020 in our district. This peritonitis rate was higher than recommended by ISPD, which may hamper generalization of the results to centers with lower peritonitis rates. In addition, 95% of the home HD patients had a fistula or graft which probably contributes to their low risk of severe infections.

Our study provides new knowledge about risk of infections in the different home dialysis modalities. Accurate information about both advantages and disadvantages of the various dialysis types is important when choosing dialysis modality. PD-related infections have been identified as a major problem among PD patients and it has been considered as one of the most important outcomes in studies on PD patients [35]. We show that home HD patients had a clearly lower risk of severe infections overall, even if their risk of bloodstream infection was higher. Notably, of the 11 bloodstream infection episodes observed among the home HD patients, no one lead to death within 30 days. Thus, our findings would favor the choice of home HD to reduce the burden of infections. However, we have earlier shown, in the same patient cohort, that PD patients did not have higher risk of death. In this study, we observed an increased mortality rate of severe infections among CAPD patients compared to home HD patients, but this comparison was unadjusted and the confidence intervals were broad, and the observation will require validation in a larger patient cohort. Many aspects need to be considered when selecting home dialysis modality. Home HD is technically more difficult, and not possible for all patients for whom PD is an option. Based on our results, the infection burden was similar for home HD and PD patients, and even lower for APD patients, when peritonitis

was excluded. It would be important to know more about risk factors for peritonitis, so that especially patients at high risk of peritonitis could be advised to choose home HD.

To conclude, CAPD and APD patients in our study had higher risk of severe infections than home HD patients. This was explained by PD-associated peritonitis. Thus, our results emphasize the importance of peritonitis prevention measures for PD patients.

## Supporting information

**S1 Table. Characteristics of home dialysis patients.**
(DOCX)

## Acknowledgments

The study nurse Katja Henttunen is acknowledged for collecting data from patient files into the study database.

## Author Contributions

**Conceptualization:** Jaakko Helve, Mari Kanerva, Eero Honkanen, Virpi Rauta, Mikko Haapio, Patrik Finne.

**Data curation:** Wisam Bitar.

**Formal analysis:** Wisam Bitar, Jaakko Helve, Patrik Finne.

**Methodology:** Wisam Bitar, Jaakko Helve, Mari Kanerva, Eero Honkanen, Virpi Rauta, Mikko Haapio, Patrik Finne.

**Resources:** Patrik Finne.

**Software:** Wisam Bitar.

**Supervision:** Jaakko Helve, Patrik Finne.

**Visualization:** Wisam Bitar.

**Writing – original draft:** Wisam Bitar.

**Writing – review & editing:** Wisam Bitar, Jaakko Helve, Mari Kanerva, Eero Honkanen, Virpi Rauta, Mikko Haapio, Patrik Finne.

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
