## [Decision Letter · Decision Letter 0]

18 Jan 2023

PONE-D-22-30726Severe infections in peritoneal dialysis and home hemodialysis patients: An inception cohort studyPLOS ONE

Dear Dr. Bitar,

Thank you for submitting your manuscript to PLOS ONE. After careful consideration, we feel that it has merit but does not fully meet PLOS ONE’s publication criteria as it currently stands. Therefore, we invite you to submit a revised version of the manuscript that addresses the points raised during the review process.

Your manuscript has been reviewed by expert referees. The critiques of the referees are shown below for your information.

1. Please provide details of statistical application to propensity score methods. Which one is used for analysis: matching or regression method? Which variables were included in the model?2. How do the authors deal with imbalanced baseline & imbalanced prognostic factors (focusing post-Rx prognosis) in non-randomized study? Please clarify explicitly.

3. How did you determine sample size and power of statistics?

4. The authors state that “The probability of the first infection episode was estimated considering death as a competing risk event…”. How do the authors deal with multiple events or district events in the multiple time-to-event with competing risk model? Fine & Grey’s proportional sub-distribution hazard model might be a good choice. Please clarify explicitly.

5. In the outcome measure’s part, please provide the definition of censoring strategy (? including multiple/recurrent event or count only the first event) and provide rationale.

6. How do the authors assess baseline Cox’s please provide both visualization and statistics? 

We look forward to receiving your revised manuscript.

Kind regards,

Wisit Kaewput, MD

Academic Editor

PLOS ONE

Journal Requirements:

2. Thank you for including your ethics statement:  "N/A".  

a. For studies reporting research involving human participants, PLOS ONE requires authors to confirm that this specific study was reviewed and approved by an institutional review board (ethics committee) before the study began. Please provide the specific name of the ethics committee/IRB that approved your study, or explain why you did not seek approval in this case.

b. Please provide additional details regarding participant consent. In the ethics statement in the Methods and online submission information, please ensure that you have specified (1) whether consent was informed and (2) what type you obtained (for instance, written or verbal, and if verbal, how it was documented and witnessed). If your study included minors, state whether you obtained consent from parents or guardians. If the need for consent was waived by the ethics committee, please include this information.

“I have read the journal's policy and the authors of this manuscript have the following competing interests: Dr. Finne has received honoraria for lectures from Baxter, AstraZeneca and Boehringer-Ingelheim, and he is a member in an advisory board of Baxter.

Dr. Rauta has received honoraria for lectures from Baxter, Fresenius and AstraZeneca, and research funding (not for this project) from Business Finland; and she is a board member in Finnish Society of Nephrology.

Dr. Honkanen has received honoraria for lectures from AstraZeneca and Fresenius Medical Care.

The findings, results, and conclusions in this report are those of the authors and are independent from the funding sources.”

Additional Editor Comments:

Your manuscript has been reviewed by expert referees. The critiques of the referees are shown below for your information.

1. Please provide details of statistical application to propensity score methods. Which one is used for analysis: matching or regression method? Which variables were included in the model? How do the authors deal with imbalanced baseline & imbalanced prognostic factors (focusing post-Rx prognosis) in non-randomized study? Please clarify explicitly.

3. How did you determine sample size and power of statistics?

4. The authors state that “The probability of the first infection episode was estimated considering death as a competing risk event…”. How do the authors deal with multiple events or district events in the multiple time-to-event with competing risk model? Fine & Grey’s proportional sub-distribution hazard model might be a good choice. Please clarify explicitly.

5. In the outcome measure’s part, please provide the definition of censoring strategy (? including multiple/recurrent event or count only the first event) and provide rationale.

6. How do the authors assess baseline Cox’s please provide both visualization and statistics?

Reviewers' comments:

Reviewer's Responses to Questions

**Comments to the Author**

1. Is the manuscript technically sound, and do the data support the conclusions?

Reviewer #1: Yes

Reviewer #2: Partly

Reviewer #3: Yes

Reviewer #4: Yes

Reviewer #5: Yes

Reviewer #6: Yes

2. Has the statistical analysis been performed appropriately and rigorously? 

Reviewer #1: Yes

Reviewer #2: Yes

Reviewer #3: No

Reviewer #4: Yes

Reviewer #5: Yes

Reviewer #6: Yes

3. Have the authors made all data underlying the findings in their manuscript fully available?

Reviewer #1: Yes

Reviewer #2: No

Reviewer #3: Yes

Reviewer #4: Yes

Reviewer #5: No

Reviewer #6: Yes

4. Is the manuscript presented in an intelligible fashion and written in standard English?

Reviewer #1: Yes

Reviewer #2: Yes

Reviewer #3: No

Reviewer #4: Yes

Reviewer #5: Yes

Reviewer #6: Yes

5. Review Comments to the Author

Reviewer #1: The authors have made an analysis of severe infections in patients starting home dialysis in their region during a 14 year period. They find that peritoneal dialysis patients (CAPD and APD) had higher risk for severe infections compared to home hemodialysis (HHD) patients, which was due to peritonitis.

Comments:

The manusdcript is very well written and easy to read. The conclusions are clear,

Methods: The analysis is based on complete data from the region, with a very clear definition of severe infection using CRP levels. As all dialysis patients in the region are taken care of by the same provider makes the study a complete analysis of the study population. That CRP is measured frequently, which makes it unlikely to miss severe infections is a further strength of the paper.

That the APD patients have very similar to the characteristics to the HHD patients is also a clear advantage.

The analysis is carefully done and very well described.

Results: Though the results are not surprising, they add valeuable data due to the completeness of patients and infection episodes from the region.

As regards HHD patients, it would be valuable to present type of hemodialysis access, in particular central lines vs. AV fistula/graft. Also, what type of cannulation was used in HHD patients with fistulas? Button-hole may be associated with a higher risk of septicemia. Did the authors analyze this? It may be difficult as access type may frequently change due to complications, but at least type of cannulation should be presented, if available.

Reviewer #2: he authors present the results of a (retrospective?) single-center study aimed to compare the risk of severe infections (defined by C-reactive protein >100 mg/L) in three cohorts of patients treated with CAPD (n=162), APD (n=229) or home HD therapy (n=145). According to their results, any group of PD suffered a higher risk of severe infections than their counterparts undergoing home HD. The burden of the difference was attributable to peritoneal infections in the first two cohorts.

The study deals an interesting, still unresolved question. The manuscript is clearly and orderly presented. The methodology is apparently well described. However, I have some concerns which should be clarified, particularly related to the way results are presented.

- The size of the sample may be insufficient to make firm conclusions, although I acknowledge that the apparent clarity of the results downplays this concern

- Patients treated with APD and home HD did not look very different, but this is not the case for CAPD patients who, for instance were markedly older and more comorbid. As the authors know (and state in the Discuccion), these marked differences may not be adequately corrected by adjustments (risk of residual bias). This limitation is relevant.

- The overall incidence of PD-related peritonitis was high. The opposite may apply for access and blood-borne infections in home HD. This performs against the external validity of the results.

- The definition criteria for severe infection are not totally clear to me, particularly at the time of presenting the Results. For instance, I have doubts on how were peritoneal infections categorized. Were all of them classified as severe? It seems that, at least, the majority were considered as such. On average, a bloodstream infection is worse than an average peritoneal infection. This question affects also the hospital admission rates. Is hospitalization a part of the protocol of management of peritonits? Only 12% of these infections demanded catheter removal, which argues against the presumed severity of these infections. The peritonitis death rate was 7,6% for CAPD and 2,5% for APD patients, which dose not suggest a particular severity either. Overall, are we talking about equally severe infections in the study groups? This question MUST be fully clarified, to evaluate the significance of the results.

- I understand that only a minority of home HD infections were blood-borne or access related (7 out of 52). What type of severe infections did these patients suffer? The number and type of considered infections should be presented for the three study groups.

- Please, state the type of vascular access used in home HD patients (day 90 could be OK). If adequate, present data on the risk of infection in patients with fistula and catheter.

- Was there a significant difference between the incidence of severe infections between CAPD and APD patients?

- In the Discussion, please, provide potential explanations for the differences observed (type of patients, dialysis access, criteria for infection severity and so…).

Reviewer #3: This is a descriptive retrospective cohort study which aimed to examine the differences in severe infections amongst home peritoneal dialysis patients and home hemodialysis patients in Finland. While the topic is important, the present submission has multiple methodological concerns.

1. The present submission is lacking a coherent hypothesis.

2. The definition of severe infection as guided by CRP is not validated.

3. The patient cohorts are two different patient populations and comparison between the 2 different groups may be in fact related to the innate difference amongst the subjects rather than the disease state or dialysis modality.

4. The interpretation of the results is misleading and can't be generalized.

5. There was little description of clinical pathway, co-interventions and modifying variables (such as patient retraining procedures and education efforts).

In summary, it is unfortunate that the present submission is unable to capitalize from the unique infrastructure to delineate the burden of infectious complications of home dialysis.

Reviewer #4: I want to congratulate the authors for this excellent work. The manuscript is well-written, has a sound methodology, and, most importantly, shows fascinating results. I believe this work is suitable for publication in PlosOne. Nevertheless, there are a few issues I would like the authors to address or clarify.

1. As I understand, only infection episodes associated with elevated CRP ≥100 mg/L were counted in this manuscript (page 5, line 99). While the authors have nicely justified the use of this cutoff, I think it would help readers perceive the severity of infection classified by this CRP cutoff better if the authors could also show data on the infection episodes with CRP <100 mg/L during the follow-up. Were there any PD-related peritonitis episodes classified as non-severe infection? Or most of the non-severe infection episodes only included local infections such as exit-site infection.

2. Continuing from comment number 1, the incidence rate for severe infection was relatively high (537 episodes per 1000 patient-year) for CAPD patients, 367 of which were peritonitis (page 11, line 183). Given the number accounted for only severe infection, this implies that the overall incidence of infection (including non-severe episodes) is even higher, which may raise concerns. Giving more information on this, perhaps in the supplementary materials, will provide readers with a better understanding of the infection situation in the context of the current study.

3. CRP is known to chronically elevate up to 10-50 mg/L in patients undergoing maintenance dialysis (Kalantar-Zadeh K. Clin Am J Soc Nephrol 2007;2(5):872-5). Several factors could explain this besides infection, such as uremic toxins or bio-incompatibility during dialysis procedures. Providing baseline CRP values before infection, if available, would be very insightful. Also, I would suggest mentioning more about the use of CRP in a specific population known to have elevated baseline CRP, like dialysis patients in the discussion (page 15, line 254).

4. Although not the main focus of the current study, the authors show in Table 3 that the incidence rate (IR) and IR ratio (IRR) of fatal infection are strikingly higher in CAPD patients compared to the other groups. I am aware that these are unadjusted, thus, not solely the effect of dialysis modality but instead attributable to other poor prognostic factors linked to CAPD, as shown in Table 1 & supplementary table 1. However, this could be misleading. At the end of the discussion, the authors also stated that PD patients from the same cohort did not have a higher risk of death. I suggest adding a brief discussion explaining the difference in the IRR of fatal infection observed in this study.

Reviewer #5: Thank you to the authors for giving the opportunity to review their article. It is an original and interesting article that compares the risk of serious infection in different home dialysis techniques.

They define severe infection by a CRP level above 100 mg, without additional bacteriological information. This is an unconventional definition but it is based on the clinical and evolutionary context detailed in the first part of the methods section. Adopting this criteria in future reseach work mightbe iterestingfor comparisons.

The abstract is a faithful summary of the main text.

It would be desirable either to add a concluding paragraph or to use the last lines of the discussion as a conclusion.

It is interesting to have separated dialysis-related infections from other episodes not related to the type of dialysis.

In the first line of Statistical analyses: to analyse the probability of the first infection episode they considered death as a competing risk; that's right but what not considerering then transfers to HD and transplantations as competing risks ?

The results were adjusted for age and gender. But they should also be adjusted for diabetes, which is more frequent in PD than in home haemodialysis.

Reviewer #6: Comparing survival , technique failure (or infection rates) between PD and HD patients is usually burdened by selection bias and residual confounding. This interesting study -based on propensity scores-tries to overcome these dificulties and provide answers about infection rates in CAPD, APD and Home Hemodialysis patients.

THe study is written in clear language and is organized nicely. The statistical analysis is the appropriate one for such kind of studies and the results are clearly stated.However , I have some comments.

1. There are no data in peritonitis area ,which correlate CRP with severity. Relevant studies usually refer to pneumonia. From my point of view, there in no need to define "severe infections" based on CRP. The authors could just state their data about CRP, hospitalizations, days of hospitalizations and infections. Moreover , I would prefer omitting "severe" from the title.

2. I missed data about kind of infections eg "non-dialysis infections " were pneumonia, urinary tract infections or something else?. There are no data about microbiology of peritonitis episodes(Gram +, Gram -). This is essential, as the authors did comment on the "higher than recommended" peritonitis incidence in their patients. So, in the discussion section, they should underline that these results apply only for this population with relatively high peritonitis incidence.

3. The authors state that 122 patients were tranferred to "in centre HD" during follow-up. Are these PD or home dialysis patients?Please comment on the reasons.

An a minor comment Ref 18 ,The study has already been published in 2022.

6. PLOS authors have the option to publish the peer review history of their article (what does this mean?). If published, this will include your full peer review and any attached files.

Reviewer #1: No

Reviewer #2: No

Reviewer #3: No

Reviewer #4: No

Reviewer #5: No

Reviewer #6: No

---

## [Author Response · Author response to Decision Letter 0]

29 Mar 2023

Additional Editor Comments:

Your manuscript has been reviewed by expert referees. The critiques of the referees are shown below for your information.

1. Please provide details of statistical application to propensity score methods. Which one is used for analysis: matching or regression method? Which variables were included in the model?

Authors’ reply: Propensity scores were calculated using logistic regression to estimate the probability of a certain treatment modality. This propensity score summarized the information from all confounders in one score and was inserted as one adjusting variable (not as a matching factor) in the Cox regression model. The methodology behind this, is described in reference 33. The explanatory variables used for the propensity scores were selected out of all variables listed in the Supplementary Table S1 using a forward stepwise procedure. The variables that were finally used to estimate the propensity scores are indicated in Supplementary Table S1. 

2. How do the authors deal with imbalanced baseline & imbalanced prognostic factors (focusing post-Rx prognosis) in non-randomized study? Please clarify explicitly.

Authors’ reply: Thank you for this important comment. We acknowledge the possible sources of bias in this non-randomized study. Unfortunately, randomization of patients into various dialysis modalities has not been successful and observational studies have to be used. The possible imbalance in baseline and prognostic factors were dealt with by calculating propensity scores that took into account a large number of potentially confounding variables. Compared to CAPD, patients on APD and home HD more often received kidney transplantation, which may cause an imbalance, but we censored for kidney transplantation in the statistical analysis. In addition, we have now performed a sensitivity analysis in which kidney transplantation was defined as a competing risk event in Fine-Gray regression, and this did not alter the results. We have added information about the Fine-Gray regression in the Methods section on page 7, lines 132–137 and mentioned the finding in the Results on page 10, lines 177–182.

3. How did you determine sample size and power of statistics?

Authors’ reply: We included the maximal available number of patients in our healthcare district who had information in the electronical patient data system, that is from 2004 onwards. The nature of the analysis in this study is exploratory, not hypothesis-driven, and therefore no power calculation was performed. In the main analysis, we found clear statistically significant differences, with rather narrow confidence intervals, between the treatment modalities. Thus, we consider the statistical power to be sufficient.

4. The authors state that “The probability of the first infection episode was estimated considering death as a competing risk event…”. How do the authors deal with multiple events or district events in the multiple time-to-event with competing risk model? Fine & Grey’s proportional sub-distribution hazard model might be a good choice. Please clarify explicitly.

Authors’ reply: In the analysis of cumulative incidence, the first infection episode was considered as the event, while death was considered as a competing event because infection episodes cannot occur after death. As infection episodes can occur (although our follow-up ended) after kidney transplantation, transfer to other dialysis modality or loss to follow, we censored for these. We decided to use Cox regression without accounting for competing events as the aim was to explore the aetiological role of potentially causal factors. We have now added a reference on the methodology of competing risk models (Noordzij et al 2013) [31], in which the choice of Cox regression is discussed. According to your suggestion, we made sensitivity analyses using Fine-Grey regression in which death, kidney transplantation and transfer to in-center HD were considered as competing events. The result did not differ markedly from that of Cox regression. We added a sentence about these sensitivity analyses to the Results section, page 10, lines 177–182.

5. In the outcome measure’s part, please provide the definition of censoring strategy (? including multiple/recurrent event or count only the first event) and provide rationale.

Authors’ reply: When calculating cumulative incidence of infection and hazard ratio of infection we included only the first infection. We have now added information about the censoring strategy into the Methods section pate 7, lines 132–137. To evaluate the censoring strategy, we made three sensitivity analyses with Fine-Gray regression to study death, kidney transplantation and transfer to in-center HD as competing risk events. These analyses did not alter the results of our main analysis. Please also see our reply to item 4. We added these sensitivity analyses to the Results section, page 10, lines 177–182.

6. How do the authors assess baseline Cox’s please provide both visualization and statistics?

Authors’ reply: We assessed the assumption that the cumulative hazards of infection episodes are proportional between the major study groups of CAPD, APD, and home HD. We evaluated the proportional hazards assumption graphically by using the Figure shown below. We have now asked our statistician to test the proportional hazards assumption statistically using the Stata software (estat phtest) giving a P Value of 0.469, indicating that the proportionality between the hazards do not change during the follow-up time. We have added a description of this methodology to the Methods, page 7, lines 140–143 and a notion to the results, Page 10, lines 177–178.

5. Review Comments to the Author

Reviewer #1: The authors have made an analysis of severe infections in patients starting home dialysis in their region during a 14 year period. They find that peritoneal dialysis patients (CAPD and APD) had higher risk for severe infections compared to home hemodialysis (HHD) patients, which was due to peritonitis.

Comments:

The manusdcript is very well written and easy to read. The conclusions are clear,

Methods: The analysis is based on complete data from the region, with a very clear definition of severe infection using CRP levels. As all dialysis patients in the region are taken care of by the same provider makes the study a complete analysis of the study population. That CRP is measured frequently, which makes it unlikely to miss severe infections is a further strength of the paper.

That the APD patients have very similar to the characteristics to the HHD patients is also a clear advantage.

The analysis is carefully done and very well described.

Results: Though the results are not surprising, they add valeuable data due to the completeness of patients and infection episodes from the region.

Authors’ reply: We thank the reviewer for the positive comments.

As regards HHD patients, it would be valuable to present type of hemodialysis access, in particular central lines vs. AV fistula/graft. Also, what type of cannulation was used in HHD patients with fistulas? Button-hole may be associated with a higher risk of septicemia. Did the authors analyze this? It may be difficult as access type may frequently change due to complications, but at least type of cannulation should be presented, if available.

Authors’ reply: At the end of the years 2004-2019, on average 91% of the home hemodialysis had a fistula and 4% had a graft, while only 5% had a tunneled HD catheter. All patients in the unit are trained to use the button-hole technique. We added a notion about these things in the Methods, page4 and 5, lines 86–89.

Reviewer #2: The authors present the results of a (retrospective?) single-center study aimed to compare the risk of severe infections (defined by C-reactive protein >100 mg/L) in three cohorts of patients treated with CAPD (n=162), APD (n=229) or home HD therapy (n=145). According to their results, any group of PD suffered a higher risk of severe infections than their counterparts undergoing home HD. The burden of the difference was attributable to peritoneal infections in the first two cohorts.

The study deals an interesting, still unresolved question. The manuscript is clearly and orderly presented. The methodology is apparently well described. However, I have some concerns which should be clarified, particularly related to the way results are presented.

- The size of the sample may be insufficient to make firm conclusions, although I acknowledge that the apparent clarity of the results downplays this concern

Authors’ reply: Thank you for this comment. We included the maximal available number of patients in our healthcare district who had information in the electronical patient data system, that is from 2004 onwards. In the main analysis, we found clear statistically significant differences, with rather narrow confidence intervals, between the treatment modalities. Thus, we consider the statistical power to be sufficient.

- Patients treated with APD and home HD did not look very different, but this is not the case for CAPD patients who, for instance were markedly older and more comorbid. As the authors know (and state in the Discuccion), these marked differences may not be adequately corrected by adjustments (risk of residual bias). This limitation is relevant.

Authors’ reply: This is an important and relevant comment. The problem of confounding is difficult to overcome in an observational study. Unfortunately, randomization of patients into various dialysis modalities has not been successful and observational studies have to be used. We dealt with the imbalance in baseline factors by calculating propensity scores that took into account a large number of potentially confounding variables including age, a large number of comorbidities and many other variables. A strength of our study is the large number of potential confounders that were taken into account in the analyses. These potential confounders have been listed in Supplementary Table 1.

- The overall incidence of PD-related peritonitis was high. The opposite may apply for access and blood-borne infections in home HD. This performs against the external validity of the results.

Authors’ reply: The concern of the reviewer is justified. The peritonitis rate was indeed slightly higher than the ISPD target at the time. But, even if the peritonitis rate would have been 20% lower, the infection rate among PD patients would have been higher than among the home HD patients. However, our results will not be generalizable to centers with considerably lower peritonitis rates. We added a sentence about this limitation in the Discussion, page 18, lines 301–304.

- The definition criteria for severe infection are not totally clear to me, particularly at the time of presenting the Results. For instance, I have doubts on how were peritoneal infections categorized. Were all of them classified as severe? It seems that, at least, the majority were considered as such. 

Authors’ reply: We used an elevated CRP > 100 mg/l as the method to identify all patients with a suspicion of a significant infection and we also used this cutoff of CRP to define a severe infection. We checked all patient files carefully and excluded patients who had an elevated CRP not caused by infections. Thus, peritonitis episodes with a CRP lower than 100 mg/l were not included, and all infection episodes included had a CRP of 100 mg or higher and they were classified as severe based on the elevated CRP. 

On average, a bloodstream infection is worse than an average peritoneal infection. This question affects also the hospital admission rates. Is hospitalization a part of the protocol of management of peritonits? Only 12% of these infections demanded catheter removal, which argues against the presumed severity of these infections. The peritonitis death rate was 7,6% for CAPD and 2,5% for APD patients, which dose not suggest a particular severity either. Overall, are we talking about equally severe infections in the study groups? This question MUST be fully clarified, to evaluate the significance of the results.

Authors’ reply: We understand the concern of the reviewer. A common way to describe infection severity is the risk of death connected to the infection. We observed 52 severe infection episodes among the home hemodialysis patients, of which 11 were caused by sepsis. One home hemodialysis patient died within 30 days after an infection episode, but no one died after a sepsis episode. With regard to severity this is comparable to peritoneal dialysis with 355 (197 among CAPD and 158 among APD patients) severe infection episodes that were followed by 19 deaths within 30 days. These numbers can be found in Table 3. To clarify, we have now added this sentence: “Notably, of the 11 bloodstream infection episodes observed among the home HD patients, no one lead to death within 30 days.” to the Discussion, page 18, line 312–313.

Our unit does not have a protocol that requires hospitalization of peritonitis patients, although most peritonitis patients are hospitalized. We have mentioned in the Discussion (lines 302–304) that the hospitalization rates due to peritonitis may not be generalized to other units.

- I understand that only a minority of home HD infections were blood-borne or access related (7 out of 52). What type of severe infections did these patients suffer? The number and type of considered infections should be presented for the three study groups.

Authors’ reply: Of the seven dialysis-related infection episodes observed among home HD patients, 5 were bloodstream infections and 2 were other access-related infections, and this is shown in Table 3. To further clarify, we have now added a Table (Table 4), that describes all the non-dialysis-related infections in the three study groups, and we have added a sentence in the Results, page 14 lines 226–227. We have also added a Table (Table 5) that describes the causative organisms of the peritonitis episodes with an accompanying sentence in the Results, page 14, lines 229–230.

- Please, state the type of vascular access used in home HD patients (day 90 could be OK). If adequate, present data on the risk of infection in patients with fistula and catheter.

Authors’ reply: We have data on the distribution of vascular access types at the end of each year for the cohort of home HD patients. On average, at the end of the years 2004 to 2019, 91% of the home HD patients had a fistula, 4% had a graft and 5% had a tunneled hemodialysis catheter. Due to the low number of catheters, adequate estimates of infection risks could not be estimated according to the type of vascular access. We have now added a sentence to the Methods section about the types of vascular access among the home HD patients, page 5, lines 88–89. 

- Was there a significant difference between the incidence of severe infections between CAPD and APD patients?

Authors’ reply: The comparison CAPD and APD was not the aim of study and therefore this comparison was not done. We have now performed an additional analysis of the comparison of CAPD and APD showing a hazard ratio of 1.3 (95% CI 0.94-1.9 for CAPD vs APD when adjusting for propensity scores). We added a sentence in the Results section about this, Page 10, lines 182–184. 

- In the Discussion, please, provide potential explanations for the differences observed (type of patients, dialysis access, criteria for infection severity and so…).

Authors’ reply: We have now added a notion in the Discussion about the vascular access: “In addition, 95% of the home HD patients had a fistula or graft which probably contributes to their low risk of severe infections”, page 18, lines 304–305. Regarding the criteria for infection severity, these have been described in the Methods section under the heading “Identification of severe infections” and in the fourth paragraph of the Discussion. One important explanation for the difference in infection risk between PD and home HD patients is the rather high rate of peritonitis. We have now added to the Discussion: “This peritonitis rate was higher than recommended by ISPD, which may hamper generalization of the results to centers with lower peritonitis rates.”, page 18, lines 302–304.

Reviewer #3: This is a descriptive retrospective cohort study which aimed to examine the differences in severe infections amongst home peritoneal dialysis patients and home hemodialysis patients in Finland. While the topic is important, the present submission has multiple methodological concerns.

Authors’ reply: Thank you for this comment. We have made several additional analyses according to the suggestions of the editor and all reviewers in order to clarify the methodological concerns.

1. The present submission is lacking a coherent hypothesis.

Authors’ reply: The reviewer is right. The nature of our analysis is exploratory, not hypothesis-driven, as we had no prior idea about which type of dialysis is connected to increased risk of severe infections. Still, we think that the aim of the study is highly relevant as infections cause a large proportion of dialysis patients’ deaths.

2. The definition of severe infection as guided by CRP is not validated.

Authors’ reply: The reviewer is right. To use CRP as a definition of severity of dialysis patients’ infections will need to be validated in future studies. In Finland, CRP is always measured when a severe infection is suspected, and therefor the CRP definition was very useful in order to identify all possibly severe infections. For all patients with an elevated CRP, we further ensured from the patient files that CRP was elevated due to an infection. Although not shown among home dialysis patients, earlier studies have indicated that high CRP correlates with severity of infections (references 27-29), and infection mortality in hemodialysis patients (reference 30). Our study gives evidence that CRP level is connected to infection severity also among home dialysis patients as CRP level associated with longer hospitalization and higher need for intensive care, and in case of peritonitis, with removal of PD catheter. 

3. The patient cohorts are two different patient populations and comparison between the 2 different groups may be in fact related to the innate difference amongst the subjects rather than the disease state or dialysis modality.

Authors’ reply: The reviewer is right. However, this is the case with virtually all studies that compare groups of patients according to dialysis modality. Unfortunately, randomization of patients into various dialysis modalities has not been successful and observational studies have to be used. We acknowledge the possible sources of bias in this non-randomized study. The possible imbalance in baseline and prognostic factors were dealt with by calculating propensity scores that took into account a large number of potentially confounding variables. All variables adjusted for in the propensity score analysis have been listed in Supplementary Table S1.

4. The interpretation of the results is misleading and can't be generalized.

Authors’ reply: In the Discussion, in the paragraph about limitations of the study we have added two sentences regarding generalization of our results, the other one about the rather high peritonitis rate in our unit, and the other one about the high proportion of fistulas and grafts among the home HD patients (page 18, lines 302–305). It is important that differences between our unit and other units are considered when evaluating generalizability of our results.

5. There was little description of clinical pathway, co-interventions and modifying variables (such as patient retraining procedures and education efforts).

Authors’ reply: In the methods section we have described the process of dialysis modality selection and mentioned the “home-first” policy of our unit. In connection to the description of our peritonitis prevention protocol, we have now added a sentence about patient training to the Methods section, page 4, lines 82–83. We have also added information about the training of patients to home HD, proportion of fistulas, and cannulation techniques to the Methods section (page 4–5, lines 86–89). 

In summary, it is unfortunate that the present submission is unable to capitalize from the unique infrastructure to delineate the burden of infectious complications of home dialysis.

Authors’ reply: We thank this reviewer, all other reviewers and the editor for the excellent comments. Thanks to your suggestions, we have done a large number of changes and additions, and we think that the manuscript has now improved.

Reviewer #4: I want to congratulate the authors for this excellent work. The manuscript is well-written, has a sound methodology, and, most importantly, shows fascinating results. I believe this work is suitable for publication in PlosOne. Nevertheless, there are a few issues I would like the authors to address or clarify.

1. As I understand, only infection episodes associated with elevated CRP ≥100 mg/L were counted in this manuscript (page 5, line 99). While the authors have nicely justified the use of this cutoff, I think it would help readers perceive the severity of infection classified by this CRP cutoff better if the authors could also show data on the infection episodes with CRP <100 mg/L during the follow-up. Were there any PD-related peritonitis episodes classified as non-severe infection? Or most of the non-severe infection episodes only included local infections such as exit-site infection.

Authors’ reply: Thank you for this important comment. We used CRP as the method to identify all potentially severe infections, and if CRP was elevated, the infection was further manually confirmed from the patient files by our study nurse. Unfortunately, due to this design, we do not have information about the infections which did not elevate plasma CRP to 100 mg/l or higher. However, we were able to show that the CRP level was connected to infection severity as a level of 200 mg/l or higher correlated with worse outcomes than a level of 100-199 mg/l. In the methods section we have mentioned the overall peritonitis rate of 530 per 1000 patient-years in our district which is higher than the rate of peritonitis with a CRP of 100 or higher reported in this study. Based on this difference, and our clinical experience, part of the peritonitis episodes are not accompanied by a CRP increase higher than 100. We have now added a notion highlighting this in the Discussion, lines 299–302. 

2. Continuing from comment number 1, the incidence rate for severe infection was relatively high (537 episodes per 1000 patient-year) for CAPD patients, 367 of which were peritonitis (page 11, line 183). Given the number accounted for only severe infection, this implies that the overall incidence of infection (including non-severe episodes) is even higher, which may raise concerns. Giving more information on this, perhaps in the supplementary materials, will provide readers with a better understanding of the infection situation in the context of the current study.

Authors’ reply: The reviewer is right. The overall infection rate is higher than the rate of infections with elevated CRP. We added a sentence about this to the Discussion, please see our explanation in our reply to item number 1. 

3. CRP is known to chronically elevate up to 10-50 mg/L in patients undergoing maintenance dialysis (Kalantar-Zadeh K. Clin Am J Soc Nephrol 2007;2(5):872-5). Several factors could explain this besides infection, such as uremic toxins or bio-incompatibility during dialysis procedures. Providing baseline CRP values before infection, if available, would be very insightful. Also, I would suggest mentioning more about the use of CRP in a specific population known to have elevated baseline CRP, like dialysis patients in the discussion (page 15, line 254).

Authors’ reply: To address this important comment, we checked the patients’ first CRP values at the time of dialysis initiation. The median CRP was 4 mg/l, with an interquartile range of 3-7 mg/l. We also checked the 90-percentile which was 15 mg/l, which means that 90% of the patients had baseline CRP lower than 15 mg/l. We checked separately the baseline CRP of patients who further developed a severe infection with elevated CRP: median 5 mg/l, interquartile range 3-8 mg/l, 90-percentile 16 mg/l. We added a notion about the baseline CRP to the Results section, page 8, lines 163–166.

Regarding the use of CRP among dialysis patients, our experience is that it is used in almost the same way as in other populations, although we have not found references to this in the literature. It must be noted that some dialysis patients have a continuously elevated CRP level due to chronic inflammation, which has to be taken into account when interpreting the CRP value. However, as we show here, the large majority of the patients in the study population had a CRP lower than 15 mg/l. What is more, for all patients with an elevated CRP value of 100 mg/l or higher, we checked from patient files if the CRP was elevated due to an infection.

4. Although not the main focus of the current study, the authors show in Table 3 that the incidence rate (IR) and IR ratio (IRR) of fatal infection are strikingly higher in CAPD patients compared to the other groups. I am aware that these are unadjusted, thus, not solely the effect of dialysis modality but instead attributable to other poor prognostic factors linked to CAPD, as shown in Table 1 & supplementary table 1. However, this could be misleading. At the end of the discussion, the authors also stated that PD patients from the same cohort did not have a higher risk of death. I suggest adding a brief discussion explaining the difference in the IRR of fatal infection observed in this study.

Authors’ reply: According to the reviewer’s suggestion we have now added a sentence to the discussion to address the higher mortality rate connected to severe infections among CAPD patients and the difficulty of interpreting this finding, lines 315-318.

Reviewer #5: Thank you to the authors for giving the opportunity to review their article. It is an original and interesting article that compares the risk of serious infection in different home dialysis techniques.

They define severe infection by a CRP level above 100 mg, without additional bacteriological information. This is an unconventional definition but it is based on the clinical and evolutionary context detailed in the first part of the methods section. Adopting this criteria in future reseach work mightbe iterestingfor comparisons.

The abstract is a faithful summary of the main text.

It would be desirable either to add a concluding paragraph or to use the last lines of the discussion as a conclusion.

Authors’ reply: We thank this reviewer for the constructive comments. We have now added a brief concluding paragraph at the end of the discussion, lines 324-326.

It is interesting to have separated dialysis-related infections from other episodes not related to the type of dialysis.

In the first line of Statistical analyses: to analyse the probability of the first infection episode they considered death as a competing risk; that's right but what not considerering then transfers to HD and transplantations as competing risks ?

Authors’ reply: To evaluate our censoring strategy, we have now performed sensitivity analyses using Fine-Grey regression in which death, kidney transplantation and transfer to in-center HD were considered as competing risk events. The result did not differ markedly from that of Cox regression. We added a sentence about these sensitivity analyses to the Results section, page 10, lines 177–182. We also refer to our reply to the Editor’s items 2 and 4.

The results were adjusted for age and gender. But they should also be adjusted for diabetes, which is more frequent in PD than in home haemodialysis.

Authors’ reply: We used propensity score adjustment to allow adjustment for a large number of potential confounders, including diabetes. Please see our replies to the Editor’s items 1 and 2. Type 2 diabetes as a comorbidity was included in the propensity score for the comparison of CAPD with home HD (see Supplementary Table S1). When comparing APD with home HD, prevalence of type 2 diabetes was similar in both groups (and thus type 2 diabetes could not be a confounder here) and it was left out from the propensity score in the statistical variable selection. On the other hand, the variable “primary kidney disease”, which contains information on both type 1 and type 2 diabetes, remained statistically significant and was included in all propensity scores. 

Reviewer #6: Comparing survival , technique failure (or infection rates) between PD and HD patients is usually burdened by selection bias and residual confounding. This interesting study -based on propensity scores-tries to overcome these dificulties and provide answers about infection rates in CAPD, APD and Home Hemodialysis patients.

THe study is written in clear language and is organized nicely. The statistical analysis is the appropriate one for such kind of studies and the results are clearly stated.However , I have some comments.

1. There are no data in peritonitis area ,which correlate CRP with severity. Relevant studies usually refer to pneumonia. From my point of view, there in no need to define "severe infections" based on CRP. The authors could just state their data about CRP, hospitalizations, days of hospitalizations and infections. Moreover , I would prefer omitting "severe" from the title.

Authors’ reply: We thank the reviewer for this comment. It is correct that there is no earlier data in the peritonitis area about the correlation of CRP with severity. We think that removing the word “severe” would be misleading as our study did not include all possible infections occurring in the study population. Our definition of severity based on CRP has been explained in the article. We think it can be argued that the observed infection episodes were severe as 88% of them led to hospitalization. We also observed an association between CRP level and severity: a higher CRP value correlated with longer hospital stay, more need for intensive care, and larger need for PD catheter removal. Although not shown among home dialysis patients, earlier studies have indicated that high CRP correlates with severity of infections (references 27-29), and infection mortality in hemodialysis patients (reference 30). Taken together, this supports defining infections with an elevated CRP as severe.

2. I missed data about kind of infections eg "non-dialysis infections " were pneumonia, urinary tract infections or something else?. There are no data about microbiology of peritonitis episodes(Gram +, Gram -). This is essential, as the authors did comment on the "higher than recommended" peritonitis incidence in their patients. So, in the discussion section, they should underline that these results apply only for this population with relatively high peritonitis incidence.

Authors’ reply: This is an important comment. We have now added two tables, Table 4 and Table 5. Table 4 presents the types of non-dialysis-related infections in the three study groups, and we have added a sentence about this in the Results, page 14 lines 226–227. Table 5 describes the causative organisms of the peritonitis episodes, and we added a sentence about this in the Results, page 14, lines 229–230. We have also added a sentence to the discussion about the high peritonitis rate and mentioned that it may hamper generalization of the results, page 18, lines 302-304.

3. The authors state that 122 patients were tranferred to "in centre HD" during follow-up. Are these PD or home dialysis patients?Please comment on the reasons.

Authors’ reply: Of the 122 patients who transferred to in-center HD, 58 were on CAPD, 49 on APD and 15 on home HD. For 35% of the PD patients, the reason for transferring to in-center HD was peritonitis. This has now been mentioned in the Methods section, page 6, lines 127-128. 

An a minor comment Ref 18 ,The study has already been published in 2022.

Authors’ reply: Thank you for the careful reading. We have corrected this.

6. PLOS authors have the option to publish the peer review history of their article (what does this mean?). If published, this will include your full peer review and any attached files.

Do you want your identity to be public for this peer review? For information about this choice, including consent withdrawal, please see our Privacy Policy.

Reviewer #1: No

Reviewer #2: No

Reviewer #3: No

Reviewer #4: No

Reviewer #5: No

Reviewer #6: No

---

## [Decision Letter · Decision Letter 1]

7 May 2023

PONE-D-22-30726R1Severe infections in peritoneal dialysis and home hemodialysis patients: An inception cohort studyPLOS ONE

Dear Dr. Bitar,

Thank you for submitting your manuscript to PLOS ONE. After careful consideration, we feel that it has merit but does not fully meet PLOS ONE’s publication criteria as it currently stands. Therefore, we invite you to submit a revised version of the manuscript that addresses the points raised during the review process.

I really appreciate your hard work and tremendous improvement. However, there are some concerns that need to be addressed, explicit. 1. The authors should manage confounders with double adjustment for propensity score (focusing pre Rx indication/contraindication)+prognostic score (focusing post Rx prognosis).2. Please provide your justification with clear reasoning/supporting evidence for comments from the Reviewer 2 and 3. 

We look forward to receiving your revised manuscript.

Kind regards,

Wisit Kaewput, MD, FACPT

Academic Editor

PLOS ONE

Journal Requirements:

**Additional Editor Comments:**

The authors addressed all my concerns properly. I have no additional comments.

Reviewers' comments:

Reviewer's Responses to Questions

**Comments to the Author**

1. If the authors have adequately addressed your comments raised in a previous round of review and you feel that this manuscript is now acceptable for publication, you may indicate that here to bypass the “Comments to the Author” section, enter your conflict of interest statement in the “Confidential to Editor” section, and submit your "Accept" recommendation.

Reviewer #1: All comments have been addressed

Reviewer #2: All comments have been addressed

Reviewer #3: (No Response)

Reviewer #4: All comments have been addressed

Reviewer #5: All comments have been addressed

Reviewer #6: All comments have been addressed

2. Is the manuscript technically sound, and do the data support the conclusions?

Reviewer #1: Yes

Reviewer #2: Yes

Reviewer #3: No

Reviewer #4: Yes

Reviewer #5: Yes

Reviewer #6: Yes

3. Has the statistical analysis been performed appropriately and rigorously? 

Reviewer #1: Yes

Reviewer #2: I Don't Know

Reviewer #3: No

Reviewer #4: Yes

Reviewer #5: Yes

Reviewer #6: N/A

4. Have the authors made all data underlying the findings in their manuscript fully available?

Reviewer #1: Yes

Reviewer #2: Yes

Reviewer #3: No

Reviewer #4: Yes

Reviewer #5: Yes

Reviewer #6: Yes

5. Is the manuscript presented in an intelligible fashion and written in standard English?

Reviewer #1: Yes

Reviewer #2: Yes

Reviewer #3: Yes

Reviewer #4: No

Reviewer #5: Yes

Reviewer #6: Yes

6. Review Comments to the Author

Reviewer #1: The authors have carefully replied to the comments by the reviewers and adequately revised their manuscript. I have no further suggestions.

Reviewer #2: The authors have fully and satisfactorily addressed my main concerns. I still have a feeling that the study compares therapies which are very difficult to match, but I appreciate that a maximum effort has been made in the statistical analysis to correct for biases, and one cannot expect to go further under a retrospective design. The explanations given in the Discussion concerning the limitations of the study seem sufficient to me.

Reviewer #3: This is an observational cohort which aimed to compare infection complications between home hemodialysis versus home peritoneal dialysis.

1. The present home dialysis population are divergent. Even the best statistical methods cannot adjust the dichotomous distribution of the patient populations.

2. The use of CRP is not validated and is contrary to ISPD peritonitis guidelines.

3. The conclusion based on the present data is unjustified.

Reviewer #4: The authors have done an excellent job addressing all my comments. I honestly think the current version is ready for publication.

Reviewer #5: I have carefully read the entire revised version and compared it to the previous version and my previous comments. All my questions have been adequately answered. Therefore, I have no additional comments.

Reviewer #6: My comments were addressed by the authors...................................................................................................

7. PLOS authors have the option to publish the peer review history of their article (what does this mean?). If published, this will include your full peer review and any attached files.

Reviewer #1: No

Reviewer #2: No

Reviewer #3: No

Reviewer #4: No

Reviewer #5: No

Reviewer #6: No

---

## [Author Response · Author response to Decision Letter 1]

11 May 2023

Rebuttal letter

We thank the academic editor and all reviewers for your great effort, which has really helped us to improve this article.

Comments from Academic Editor

I really appreciate your hard work and tremendous improvement. However, there are some concerns that need to be addressed, explicit. 

1. The authors should manage confounders with double adjustment for propensity score (focusing pre Rx indication/contraindication)+prognostic score (focusing post Rx prognosis).

2. Please provide your justification with clear reasoning/supporting evidence for comments from the Reviewer 2 and 3. 

Authors’ response: 

Thank you very much for your helpful comments and for inviting us to submit a revised version.

1. In order to confound a relation between exposure (here dialysis modality) and outcome (infection event) a variable has to be associated with both the exposure and the outcome. If it is not associated with one of these two, confounding is not possible. The propensity score was constructed from variables (all measured before start of dialysis) that were independently associated with the exposure and it included variables related to indication and contraindications for the dialysis modalities. Such variables were primary renal disease (especially polycystic degeneration less common among PD patients), obesity (favored choice of home HD), visual problems (favored choice of PD), compliance problems (favored choice of PD), high age (favored choice of CAPD), need for support in daily activities (favored CAPD). All these variables were included in the propensity score. Other indication variables were evaluated as well (see list of all variables in Supplementary S1 Table), but not included as they were not significant. Thus, the currently used propensity score already focuses on pre-Rx indication/contraindication variables. In addition, the propensity scores include some other variables that may be connected to prognosis (that is risk of infection episode). These are for example: malignancy, hypertension, atrial fibrillation, albumin, CRP, and hemoglobin.

According to your suggestion, we have now added to the adjustment prognostic variables that were not included in the propensity score. To identify such prognostic variables, we included all variables in Supplementary S1 Table that had not been included in the propensity score in a stepwise forward Cox regression analysis. In this way, we were able to identify the prognostic variables that had not been included in the propensity score. We then performed a sensitivity analysis with adjustment for both the propensity score and the prognostic variables. The methodology of this analysis has been described on lines 154–159. We added the result of this analysis on lines 190 and 193–196. This analysis did not change the main result of our study.

2. Please see our answers to reviewers 2 and 3.

Reviewer #2

Reviewer’s comment: The authors have fully and satisfactorily addressed my main concerns. I still have a feeling that the study compares therapies which are very difficult to match, but I appreciate that a maximum effort has been made in the statistical analysis to correct for biases, and one cannot expect to go further under a retrospective design. The explanations given in the Discussion concerning the limitations of the study seem sufficient to me.

Authors’ response: Thank you for your comment. You are right. Observational studies always have a risk of confounding and bias. As it has not been possible to randomize patients into dialysis modalities, evidence has to be sought from observational studies. At least three issues support that there is a causal relation between dialysis modality and infection burden if looking at the Bradford Hill criteria for causality:

1) Strength in effect size with a hazard ratio of >2

2) Temporality: before start of dialysis there was no difference in rate of infection episodes between patients who later started PD or home HD

3) Plausibility: It is plausible that PD patients have larger infection burden than home HD patients because of their peritonitis risk. Even if randomizing patients to PD or home HD it is likely that the patients randomized to PD would have higher peritonitis risk than those randomized to home HD.

Reviewer #3

Reviewer’s comments: This is an observational cohort which aimed to compare infection complications between home hemodialysis versus home peritoneal dialysis.

1. The present home dialysis population are divergent. Even the best statistical methods cannot adjust the dichotomous distribution of the patient populations.

2. The use of CRP is not validated and is contrary to ISPD peritonitis guidelines.

3. The conclusion based on the present data is unjustified.

Authors’ response:

Thank you for your comments. 

1) The reviewer is right. Observational studies always have a risk of confounding and bias. Unfortunately, it has not been possible to randomize patients into dialysis modalities. Thus, evidence has to be sought from observational studies. A number of issues support that there is a causal relation between dialysis modality and infection burden in our study according to the Bradford Hill criteria for causality. Please see our answer to Reviewer 2.

2) Thank you for this question. We have mentioned in the article that CRP has not been validated. The CRP definition was especially useful in the current study in order to identify all possibly severe infections, because in Finland, CRP is always measured when a severe infection is suspected. For all patients with an elevated CRP, we further ensured from the patient files that CRP was elevated due to an infection. Although not shown among home dialysis patients, earlier studies have indicated that high CRP correlates with severity of infections, and infection mortality in hemodialysis patients. Our study gives evidence that CRP level is connected to infection severity also among home dialysis patients as CRP level associated with longer hospitalization and higher need for intensive care, and in case of peritonitis, with removal of PD catheter.

The aim of our study was not to study peritonitis specifically, but all types of severe infection episodes occurring among home dialysis patients. The peritonitis rate (and the rate of other infections) would have been higher if not restricting to infections with a CRP higher than 100 mg/l. It is correct that CRP is not mentioned in the ISPD guidelines. We used CRP to identify patients with various types of severe infections in this retrospective cohort study. However, we do not suggest that CRP should be used for clinical identification of peritonitis among PD patients.

3) Our conclusion is: “To conclude, CAPD and APD patients in our study had higher risk of severe infections than home HD patients. This was explained by PD-associated peritonitis. Thus, our results emphasize the importance of peritonitis prevention measures for PD patients.” We refer to our explanations to the editor and both reviewers. We think this conclusion is justified based on the data presented.

---

## [Editor Report · Decision Letter 2]

19 May 2023

Severe infections in peritoneal dialysis and home hemodialysis patients: An inception cohort study

PONE-D-22-30726R2

Dear Dr. Bitar,

We’re pleased to inform you that your manuscript has been judged scientifically suitable for publication and will be formally accepted for publication once it meets all outstanding technical requirements.

Kind regards,

Wisit Kaewput, MD, FRCPT

Academic Editor

PLOS ONE

Additional Editor Comments (optional):

In the revised manuscript, the authors convincingly enough replied to my previous comments and properly improved the manuscript. The manuscript contains now all information. Overall the manuscript reads well, has clarity, and communicates the work of the authors. In my opinion this manuscript is suitable for publication in PLOS ONE.
---

## [Editor Report · Acceptance letter]

5 Jun 2023

PONE-D-22-30726R2 

Severe infections in peritoneal dialysis and home hemodialysis patients: An inception cohort study 

Dear Dr. Bitar:

I'm pleased to inform you that your manuscript has been deemed suitable for publication in PLOS ONE. Congratulations! Your manuscript is now with our production department. 

Kind regards, 

on behalf of

Dr. Wisit Kaewput 

Academic Editor

PLOS ONE